# Synthesis, Characterization, and Application of Dichloride (5,10,15,20-Tetraphenylporphyrinato) Antimony Functionalized Pectin Biopolymer to Methylene Blue Adsorption

**DOI:** 10.3390/polym15041030

**Published:** 2023-02-19

**Authors:** Raoudha Soury, Munirah Sulaiman Othman Alhar, Mahjoub Jabli

**Affiliations:** 1Department of Chemistry, College of Science, University of Hail, Ha’il 81451, Saudi Arabia; 2Department of Chemistry, College of Science Al-Zulfi, Majmaah University, Al-Majmaah 11952, Saudi Arabia

**Keywords:** Sb(TPP)Cl_2_, pectin, adsorption, methylene blue, Freundlich

## Abstract

In this work, pectin biopolymers were functionalized with dichloride (5,10,15,20-tetraphenylporphyrinato) antimony [Sb(TPP)Cl_2_] at various compositions (0.5%, 1%, and 2%). The prepared compounds were characterized with several analytical methods, including X-ray fluorescence (XRF) spectrometry, Fourier-transform infrared spectroscopy (FT-IR), electrospray ionization mass spectrometry (EIS), scanning electron microscope (SEM), X-ray diffraction (XRD), and thermogravimetric-differential thermal (TGA/DTG) analysis. The XRF technique evidenced the presence of Sb metal in the composite beads. FT-IR suggested that the interaction between pectin and the [Sb(TPP)Cl_2_] complex was assured by inter- and intramolecular C-H⋯O, C-H⋯Cl hydrogen bonds and weak C–H⋯Cg π interactions (Cg is the centroid of the pyrrole and phenyl rings). The morphological features of the prepared polymeric beads were affected by the addition of [Sb(TPP)Cl_2_] particles, and the surface became rough. The thermal residual mass for the composite beads (29%) was more important than that of plain beads (23%), which confirmed the presence of inorganic matter in the modified polymeric beads. At 20 °C, the highest adsorption amounts of methylene blue were 39 mg/g and 68 mg/g for unmodified pectin and pectin-[Sb(TPP)Cl_2_] beads, respectively. The adsorption mechanism correlated well with the kinetic equation of the second order and the isotherm of Freundlich. The prepared polymeric beads were characterized as moderate-to-good adsorbents. The calculated thermodynamic parameters demonstrated an exothermic and thermodynamically nonspontaneous mechanism.

## 1. Introduction

The elimination of synthetic dye molecules from contaminated water has become a global environmental worry because of the influences of contaminants on human health and ecosystems. The cationic methylene blue dye, which is water-soluble, has been stated as a main constituent of polluted water. This dye has been identified as carcinogenic and has massive toxic effects on marine organisms [1]. To overcome such a pollution problem, several conventional techniques were used for the adsorption of synthetic dyes from water. These methods comprise coagulation, flocculation, electrochemical reactions, oxidation, and adsorption [2,3,4]. Among these means, adsorption constitutes a common choice for water treatment owing to its simplicity, low price, and availability of many classes of adsorbents.

The proficiency of adsorption is dependent on the physicochemical properties of the used adsorbent and its reutilization ability. As a consequence, there is an important concern devoted to developing biomaterials derived from renewable sources, including cellulose, lignin, chitin, chitosan, pectin, carrageenan, etc. Indeed, the design of biopolymer-based adsorbents has attracted the attention of many researchers [5,6,7]. Pectin, which is essentially obtained from apple pomaces and citrus peels, is a safe and biocompatible polymer [8]. Pectin consists of (1-4)-D-galacturonic acid residues with a few residues of 1,2-D-rhamnose, D-galactose, and D-arabinose with variable degrees of esterification [9]. It is commonly used for gelling, stabilizing food, and thickening. However, it suffers from restricted thermal stability and mechanical characteristics [10]. To overcome these limits and improve the adsorption properties, pectin was blended with other materials [11]. As examples of recent studies, Lessa et al. [12] developed pectin/cellulose microfiber composite beads. The composite beads showed significant differences concerning some properties compared to the samples prepared without cellulose microfibers. The adsorption ability of methylene blue was well improved for pectin/cellulose microfibers composite beads. A prior study conducted by Kong et al. [13] on the development of chitosan-pectin composites in dimethyl sulfoxide versus water displayed that polymeric composites synthesized in dimethyl sulfoxide generated a covalent biopolymer structure, whereas a polyelectrolyte compound was obtained in water. The results exhibited that the covalent polymeric composites showed better adsorption of methylene blue than the polyelectrolyte complex. In another study, cellulose nanocrystals were utilized with acrylic acid and pectin to get a nanocomposite by γ-irradiation [14]. The nanocomposite was employed to adsorb methylene blue from water. Results indicated that the existence of cellulose nanocrystals in the polymeric matrix improved the adsorption characteristics and swelling of the nanocomposite.

Metallo-porphyrins are considered an interesting class of molecules, and they could be used for a wide variety of applications. They are characterized by their rigidity, great solubility in organic suspension, and photo-physical characteristics. They are also highly thermally stable. Owing to these interesting features, metallo-porphyrins have been used as catalysts in photocatalysis, opto-electronics, medicine, dye adsorption, etc. [15,16,17,18,19,20,21,22,23]. Herein, dichloride (5,10,15,20-tetraphenylporphyrinato) antimony, an organometallic compound, was immobilized onto a pectin biopolymer using a precipitation method. The resulting composite beads were analyzed using XRF, FT-IR, XRD, SEM, EIS, and TGA/DTG techniques. Further, the composites were employed to remove methylene blue from water by changing the experimental parameters, mainly [Sb(TPP)Cl_2_] mass, pH, time, concentration of dye, and bath temperature. The adsorption capacities were assessed and compared for plain and functionalized beads. Kinetic and isotherm studies were also carried out in an attempt to better understand the adsorption mechanism.

## 2. Experimental Section

### 2.1. Reagents and Materials

Pectin powders (M.W. = 30,000–100,000 g/mol, esterification degree: 63–66%) were supplied by Sigma Aldrich. Calcium chloride (CaCl_2_) was used as a precipitant agent for the biopolymer. Methylene blue was purchased from Sigma Aldrich. The aqueous solutions were realized with distilled water.

### 2.2. Synthesis of Sb(TPP)Cl_2_

The [Sb(TPP)Cl_2_] complex was prepared under an argon atmosphere following our previous work [24]. For this, tetraphenylporphyrin (H_2_TPP) (600 mg, 0.97 mmol) was dissolved in 30 mL of pyridine, and the reaction mixture was heated to 115 °C. SbCl_5_ (4.5 mg, 1.508 mmol) was added to a solution, and the mixture was refluxed for 1 h. After purification, a purple solid was obtained. The structure of [Sb(TPP)Cl_2_] is given in Figure 1. The structure was confirmed by EIS mass spectroscopy, proton NMR spectroscopy, and an elemental analyzer Anal. Calc. For [Sb(TPP)Cl_2_]: C_44_H_28_N_4_SbCl_2_ (805.1 g/mol). C, 45.32; H, 2.55; N, 4.80%. Found: C, 44.90; H, 2.65; N, 4.70%. UV/Vis [CH_2_Cl_2_, λ_max_ in nm (logξ)]: 428 (5.48), 557 (3.25), 598 (3.11). ^1^H-RMN (CDCl_3_, 300 MHz): δ (ppm) = 9.66 (S, 8H, Hβ), 8,34 (d, 8H, Ho,o’); 7,98 (d, 12H, Hm,m’,p). MS (ESI+, CH2Cl2): m/z = 805.1 [Sb(TPP)Cl_2_]^+^.

### 2.3. Synthesis of [Sb(TPP)Cl_2_] Functionalized Pectin

First, 4 g of pectin polymer was dissolved in 100 mL of distilled water and stirred for 24 h at room temperature. Then, the prepared [Sb(TPP)Cl_2_] particles were blended with the homogenous pectin solution, at different loading contents (0.5%, 1%, and 2%, solid pectin is considered as the main constituent). Solution was vigorously stirred for 4 h. After complete homogenization, the resulting solution was pumped into droplets in a precipitation bath of CaCl_2_ (10% *w*/*v*) through a needle (the diameter equals 0.2 mm). Then, the formed beads were filtered and washed with distilled water many times. Finally, pectin-[Sb(TPP)Cl_2_] beads were dried under vacuum at 40 °C for 48 h. Additionally, the synthesized virgin pectin beads were used as control samples.

### 2.4. Characterization Techniques

MicrOTOF Q Bruker instrument (University of Grenoble, France) was employed to implement mass spectra of the prepared [Sb(TPP)Cl_2_]. An XRF spectrophotometry (Qassim University, Saudi Arabia) was used to determine the relative abundance of metals and trace elements in the polymeric samples. An FT-IR System Perkin Elmer 2000 Model was used to identify the chemical groups present in such a sample structure. XRD patterns were carried out under the following conditions: X-ray Tube: Cu (1.54060 A) Voltage = 40.0 kV, current = 30.0 mA, scan range from 10 to 80 degrees, Thermogravimetric analysis (TGA/DTA) was carried out using a TGA 209F1D-0346-L (Qassim University, Saudi Arabia). Experiments were realized under nitrogen flow from 25 °C to 800 °C (heating rate = 10 °C/min). The morphological features of the polymeric beads were analyzed using a Hitachi S800-1 microscope (National School of Engineering, Monastir, Tunisia). The absorbance of methylene blue solution was measured in a quartz cell using a UV–visible spectrophotometer.

### 2.5. Dye Adsorption Study

Adsorption experiments were carried out in conical flasks containing 0.01 g of dried polymeric beads and 20 mL of methylene blue. The dye solution was stirred continuously (125 rpm). At the completion of each experiment, each sample was filtered, and the absorbance value was recorded at the high wavelength of methylene blue dye (660 nm). The experimental parameters, including time, pH, color concentration, and temperature, were studied at different ranges. The adsorbed amount of the cationic color onto unmodified pectin beads and pectin-[Sb(TPP)Cl_2_] composite was considered using Equation (1):

q_t_ = (C_0_ − C_t_) × V/m
(1)


C_0_ and C_t_ are the concentrations of dye solution (mg/L) at time = 0 and time = t, respectively, q_t_ is the adsorbed quantity of dye (mg/g) at time = t, V is the volume of the utilized dye solution during experiment (L), and m is the quantity of the polymeric beads (g).

## 3. Results

### 3.1. Electrospray Ionization Mass Spectrometry (EIS)

The structure of [Sb(TPP)Cl_2_] complex has been determined using EIS conducted in a solution of dichloromethane (C = 5 × 10^−5^ M) in the positive mode [20,21] (Figure 1). As it is observed, the peak at 805.1 m/z confirms the presence of [Sb(TPP)Cl_2_]^+^ moieties.

### 3.2. XRF Characterization

Table 1 summarizes the relative abundance of elements present in pectin beads and pectin composite beads (1%), as determined by the XRF technique. Results showed that the Sb element is present in the composite pectin beads with a relative amount of 0.475%. In the unmodified pectin beads, the relative abundances of Ca, Cl, and Al were found to be 62.742%, 25.098%, and 10.724%, respectively. Whereas in the pectin composite beads, the relative abundances of Ca, Cl, and Al were found to be 52.404%, 39.201%, and 7.841%, respectively. Such behavior evidences the formation of pectin-[Sb(TPP)Cl_2_] composite beads. Other elements, including Si, Nb, Mo, Sn, ln, Rh, and Ru, are found in small amounts.

### 3.3. FT-IR Characterization

Figure 2 provides the FT-IR spectrum of pectin powder, pectin beads, [Sb(TPP)Cl_2_], and pectin-[Sb(TPP)Cl_2_] composite beads. For [Sb(TPP)Cl_2_] complex, the C–H stretching frequency of the TPP moiety is observed in the range 3100–2915 cm^−1^. The absorption band at 1594 cm^−1^ is assigned to the stretching ν (C=W C). Peaks located in the range 1475–1433 cm^−1^ belong to the C-H bending. The absorption band at 1231 cm^−1^ is assigned to C-N stretching. The bands ascribed to the deformation mode δ(CCH) are seen in the range 1112–910 cm^−1^. The absorption band located between 889 and 700 cm^−1^ is assigned to the ν (C–C) phenyl group [25]. The FT-IR spectrum of pectin powder shows a large peak of approximately 3339 cm^−1^ that corresponds to hydroxyl stretching vibrations. The peaks at the region of 1741−1621 cm^−1^ are due to the vibration of COOCH_3_ and COO^−^ groups. The band at 1420 cm^−1^ is attributed to the OH bending groups. The band in the region 1300–1000 cm^−1^ is attributed to the C-O-C stretching groups [26,27,28]. Compared to the FT-IR spectrum of pectin beads, a slight shift and small differences were seen in the intensity of bands for the functional groups of the pectin powder. This is due to the interaction of these groups with calcium ions during chemical precipitation. After adding the dichloride (5,10,15,20-tetraphenylporphyrinato) antimony to pectin biopolymer, remarkable shifts are observed within the main functional groups suggesting the interaction between the pectin and [Sb(TPP)Cl_2_] complex assured by conventional and non-conventional interactions.

### 3.4. Morphological Characteristics (SEM)

Figure 3 provides the morphological features of pectin beads, and pectin-[Sb(TPP)Cl_2_] composite beads at different magnifications (×100 and ×1000). As it is observed, the unmodified pectin beads display a homogenous and smooth surface. However, this surface appears rough after the addition of [Sb(TPP)Cl_2_] particles. Such change indicates the formation of the pectin-[Sb(TPP)Cl_2_] composite beads. It is also suggested that the addition of [Sb(TPP)Cl_2_] to pectin beads affects clearly the morphology characteristics of the beads.

### 3.5. XRD Charcaterization

Figure 4 displays the XRD patterns of pectin beads and pectin-[Sb(TPP)Cl_2_] composite beads. As it is shown, the unmodified pectin beads reveal sharp peaks at 2θ that equal 13.6° θ, 20.3° θ, 30.1° θ, and 43° θ, which suggests the crystalline nature of pectin [29]. However, these peaks are observed at 2θ equals to 13.7° θ, 19.3° θ, and 30.2° θ. Indeed, the shifting of the position of the main peaks suggests that the pectin polymer is reacted with Sb(TPP)Cl_2_ particles.

### 3.6. Thermal Analysis (TGA/DTA)

The TGA/DTA curves of pectin beads and pectin-[Sb(TPP)Cl_2_] are presented in Figure 5. All the studied samples display several thermal events that are observed at a temperature <100 °C and correspond to the loss of adsorbed water [30,31]. During the thermal treatment of the compounds up to 600 °C, a total mass loss is equal to 77% for the pectin beads, which is distributed at many stages as follows: 10%, 39%, 11%, and 17%. This thermal decomposition is associated with a small endothermic peak at 230 °C, corresponding to the oxidative decomposition of pectin biopolymer. The second thermal event is an intense exothermic peak observed at 547 °C and could be attributed to the decomposition of the generated fragments. The total weight loss, for pectin-[Sb(TPP)Cl_2_] composite beads, is equal to 71%. This weight loss is subdivided into many stages, as observed at 8%, 40%, 12%, and 11%. Indeed, compared to the unmodified beads, we notice a thermal shift for the first and second thermal events occurring at 203 °C and 524 °C, respectively. Such a difference in the thermal events proves the chemical modification of the pectin biopolymer. After thermal decomposition, the residual mass observed for the composite beads (29%) is more important than that of the unmodified beads (23%), indicating the presence of inorganic matter.

### 3.7. Removal of Methylene Blue

#### 3.7.1. Experimental Parameters Change

In attempts to optimize the removal of the methylene blue using the synthesized polymeric beads, the following experimental conditions were investigated: [Sb(TPP)Cl_2_] content, time, initial pH value, initial dye concentration, and bath temperature. Figure 6a indicates that the adsorption quantity of methylene blue increases with the increase in the content of [Sb(TPP)Cl_2_] in the prepared polymeric beads and reaches its maximum when the content of [Sb(TPP)Cl_2_] reaches 1% (pH = 6, T = 20 °C, C_0_ = 30 mg/L, and time = 90 min). As a consequence, for further adsorption experiments, the removal of methylene blue will be evaluated and addressed for the unmodified pectin beads and pectin-[Sb(TPP)Cl_2_] (1%) composite beads. The pH of the solution can considerably influence the surface charge of both the solute and the solid studied adsorbent through either protonation or deprotonation events [32]. Indeed, methylene blue dye is easily adsorbed under alkaline conditions due to the attraction between the positive charges of methylene blue and the negative charges of the adsorbent [33]. Figure 6b shows that the favorable pH for the uptake of methylene blue onto the polymeric beads was close to six (T = 20 °C, time = 90 min, C_0_ = 30 mg/L). In fact, the phenomenon of adsorption is based on the electrostatic interaction between the carboxylic groups of pectin and the imine groups of the cationic dye molecules. When pH values are low, the surface of the adsorbent could be subjected to a protonation event, and therefore, the carboxylate groups might be transformed into carboxylic acid groups, resulting in smaller adsorbed amounts of dyes. However, when pH values increase, the carboxyl group of the pectin biopolymer is ionized, and these negative groups react with the positively charged dye molecule, resulting in high uptake capacities. The influence of the change in time on the removal of methylene blue is given in Figure 6c,d. Results show that the adsorbed dye quantity increases over time and then attains a persistent value. Indeed, this adsorption capacity is rapid during the first 30 min, after which it rises slowly and further reaches an equilibrium state at approximately 90 min. This trend suggests that, when the adsorption begins, there are many active sites available on the surface of the polymeric beads, which offer high adsorption capacities for cationic dyes. On the contrary, the adsorption sites on the surface of these beads will be saturated at the final stage of the adsorption, resulting in limited adsorption capacities.

Figure 6e displays the change in the adsorbed dye quantity against the methylene blue concentration (time = 90 min, and pH = 6). At low dye concentrations, the adsorption increases quickly. Indeed, either the concentration rise or driving power of such an adsorbate could be affected by the variation in the dye concentration throughout adsorption **[34,35]**. The available adsorption sites are enough when the dye concentration is low, and consequently, the adsorption can rapidly achieve an equilibrium state. When the concentration of methylene blue becomes high, the dye uptake becomes stable and then decreases with dye concentration. This is because the adsorption sites are no longer available. The plots also showed that the highest adsorption quantities of methylene blue are 39 mg/g and 68 mg/g at 20 °C for unmodified pectin beads and pectin composite beads, respectively. This dissimilarity in the adsorption capacities between unmodified and modified polymeric beads could be interpreted in relation to the occurrence of additional interactions between the polymeric composite beads and the cationic dye. The high adsorbed quantity of methylene blue obtained with the prepared composite beads is due to the intermolecular C–H⋯IO hydrogen bonds, intermolecular C–H⋯ICl, and weak C–H⋯Cg π interactions. The comparison of the maximum adsorption capacities of methylene blue dye with some examples of adsorbents reported in the literature showed that the studied functionalized pectin beads could be considered promising adsorbents (Table 2).

As it is also observed, the adsorption is influenced by the variation of the temperature of the bath (time = 90 min and pH = 6). The mechanism is exothermic, which indicates the weakness of some established interactions with the rise of temperature (Figure 6f,g).

#### 3.7.2. Kinetic Modeling

In this study, the kinetic models of first-order, Elovich, intraparticular diffusion (Appendix A), and second-order (Figure 7) were used to assess the relationship and the connection between time and the adsorbed quantity of methylene blue. The resultant kinetic parameters are determined and listed in Table 3. Following the summarized data, the kinetic equation of the second order demonstrates that the values of regression are equal to or higher than 0.99. We also observed that the adsorption capacities calculated using this kinetic equation were close to the experimental values. Thus, this finding proposes that the current mechanism follows well the second-order kinetic model. As a consequence, we can conclude that the kinetic experiments conducted under the current conditions were predominantly affected by a chemical process [41]. In addition, regarding the intraparticle diffusion model, the values of the calculated parameter C are higher than zero, which may indicate that the process has internal and external diffusions because of the boundary layer effects [42].

#### 3.7.3. Adsorption Isotherms and Calculation of the Thermodynamic Parameters

In the current study, the isotherms of Langmuir, Temkin, and Freundlich [43,44] were checked to assess the correspondence between the mechanism of methylene blue removal and the variation in dye concentration. The corresponding theoretical parameters and the regression coefficients are determined and recorded in Table 4. Results indicated that the lowest values of the regression coefficients were detected for the isotherms of Temkin and Langmuir (Appendix A) as compared to those of Freundlich (R^2^ ≥ 0.98) (Figure 8). This result suggests that multilayer adsorption may occur at heterogeneous interfaces [43]. The favorability of adsorption could also be evaluated through the calculated number n in Freundlich. In fact, if the value of n ranges from 2 to 10, this is good. If n changes from 1 to 2, this is moderate, and finally, if n is higher than 1, this is poor adsorption [45]. In this study, the prepared polymeric beads were considered moderately good adsorbent materials.

In this section, a thermodynamic study is carried out for the adsorption of methylene blue onto unmodified pectin beads and pectin-[Sb(TPP)Cl_2_] (1%) composite beads.

The free energy change (ΔG°) is calculated using Equations (2) and (3) [45]:

ΔG°= −RT LnK_d_
(2)


Kd = q_e_/C_e_
(3)


R (J. mol^−1^·k^−1^) represents the gas constant and T (K) symbolizes the temperature.

The values of ∆H° and ∆S° are calculated using Equation (4):

Ln K_d_ = −∆H°/RT + ∆S°/R
(4)


The slopes and intercepts of the linear plot of Ln Kd vs. (1/T) are used to deduce the values of ∆H° and ∆S° (Figure 9). The calculated parameters are given in Table 5. The adsorption of the cationic dye onto the prepared polymeric beads could be regarded as exothermic (∆H° < 0). During adsorption experiments, the disorder decreases at the interface between the solution and the adsorbent, eventually causing a well-ordered spreading of dye molecules at the active sites of adsorption (ΔS° < 0). Moreover, the mechanism could be considered as thermodynamically non-spontaneous (ΔG° < 0).

## 4. Conclusions

In this study, dichloride (5,10,15,20-tetraphenylporphyrinato) antimony was successfully immobilized onto pectin gel beads. The prepared polymeric beads were characterized and used to remove methylene blue from an aqueous solution. The FT-IR result showed remarkable shifts observed within the main functional groups suggesting the interaction between pectin and [Sb(TPP)Cl_2_] complex assured by conventional and nonconventional interaction. The surface features of the polymeric beads become rough after the addition of [Sb(TPP)Cl_2_] particles. After thermal decomposition, the residual mass observed for the composite beads (29%) is more important than that of the unmodified beads (23%), which proves the presence of inorganic matter in the modified polymeric beads. The maximum adsorption quantities of methylene blue were 39 mg/g and 68 mg/g for pectin beads and composite gel beads, respectively. The adsorption mechanism correlated well with the second-order equation (R^2^ ≥ 0.99), which suggests that the studied phenomenon was mainly chemical. The isotherm study revealed that multilayer adsorption might occur at heterogeneous interfaces. The prepared polymeric beads could be considered moderate-to-good adsorbents. The calculated thermodynamic parameters suggested that the adsorption of methylene blue on unmodified pectin beads and pectin-[Sb(TPP)Cl_2_] (1%) composite beads was nonspontaneous and exothermic. During the adsorption experiments, the disorder decreases at the interface between the solution and the adsorbent, causing a well-ordered spread of dye molecules at the active sites of adsorption.

## Data Availability

Not applicable.

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
