# Peer review of "Synthesis, Characterization, and Application of Dichloride (5,10,15,20-Tetraphenylporphyrinato) Antimony Functionalized Pectin Biopolymer to Methylene Blue Adsorption"

_polymers, 2023, doi:10.3390/polym15041030_

Round 1

Reviewer 1 Report

After reading the manuscript entitled: Synthesis, characterization, and application of dichloride(5,10,15,20-tetraphenylporphyrinato) antimony functionalized pectin
biopolymer to methylene blue adsorption, which reports the synthesis of a polymeric-based adsorbent for the removal of methylene blue dye, I found it average with respect to application and factors studied, however, the synthesis and characterization of the adsorbent are worth the publication after considering the following comments:

1- On page 2, lines 71-77 is repeating the abstract. in this part, the innovation and the main aim of the research should be clearly mentioned.

2- On page 2, line 92, Figure 1 should be scheme 1.

3- When explaining terms in any equation, you should provide the unit of each term.

4- Figure 1 shows many peaks other than the one at 805.1, what they are related to?

5- What do you mean by conventional and nonconventional interaction?

6- SEM images are not clear.

7- XRD in Figure 4 is very noisy and it is not sure if the peaks discussed are real or only noise!

8- The y-axes of Figures 5a and 5B should be remaining mass not weight loss.

9- The caption of Figure 5 must be thermal gravimetric analysis not thermal graphs.

10- On page 9, what is the meaning of 2%? is it 2 mg of adsorbent per 100 mg of mixture?

11- On Figure ^a, the maximum was achieved at 1% not as mentioned at 2%.

12- What is the y-axes of Figure 6a? is it SB content? or mass of adsorbent?

13- Add the experimental contions to the caption of Figure 6.

14- There is no Figure 6g as mentioned in the caption and in line 289 of page 11.

15- on page 16, line 375, it is mentioned that the adsorption was achieved via multilayer, which is usually attributed to physical adsorption not chemical as mentioned in the discussion. to ensure this point is accurate, it is better to calculate the activation energy.

Author Response

Reviewer: 1

After reading the manuscript entitled: Synthesis, characterization, and application of dichloride(5,10,15,20-tetraphenylporphyrinato) antimony functionalized pectin
biopolymer to methylene blue adsorption, which reports the synthesis of a polymeric-based adsorbent for the removal of methylene blue dye, I found it average with respect to application and factors studied, however, the synthesis and characterization of the adsorbent are worth the publication after considering the following comments:

1- On page 2, lines 71-77 is repeating the abstract. in this part, the innovation and the main aim of the research should be clearly mentioned.

Answer 1. As per reviewer comment, this paragraph is now modified as:

Herein, dichloride (5,10,15,20-tetraphenylporphyrinato) antimony, an organometallic compound, was immobilized onto pectin biopolymer using a precipitation method. The resulting composite beads were analyzed using XRF, FT-IR, XRD, SEM, EIS, and TGA/DTG techniques. Further, the composites were employed to remove methylene blue from water by changing the experimental parameters mainly [Sb(TPP)Cl2] mass, pH, time, concentration of dye, and bath temperature. The adsorption capacities were assessed and compared for plain and functionalized beads. Kinetic and isotherms studies were also carried out in attempt to better understand the adsorption mechanism.

2- On page 2, line 92, Figure 1 should be scheme 1.

Answer 2. As per reviewer comment, Figure 1 is corrected as scheme 1.

3- When explaining terms in any equation, you should provide the unit of each term.

Answer 3. As per reviewer comment, the corresponding units are now provided.

4- Figure 1 shows many peaks other than the one at 805.1, what they are related to?

Answer 4. [Sb(TPP)Cl2] showed an intense peak at m/z 805.1 However, for 804.1, 806.1 and 807.1 m/z are attributed to [Sb(TPP)Cl2-H+]+, [Sb(TPP)Cl2+H+]+ and [Sb(TPP)Cl2+2H+]+ moieties, respectively.

As the intense peak at 805.1 is the most important one, we have only mentioned this peak in the text.

5- What do you mean by conventional and nonconventional interaction?

Answer 5. Conventional and nonconventional interaction means: inter and intramolecular C– H…O, C–H…Cl hydrogen bonds, and weak C–H…Cg π interactions (Cg is the centroid of pyrrole and phenyl ring)

6- SEM images are not clear.

Answer 6. The images are presented as received from machine and we tried to provide better images.

7- XRD in Figure 4 is very noisy and it is not sure if the peaks discussed are real or only noise!

Answer 7. The images are presented as received from machine and we tried to provide better curve.

8- The y-axes of Figures 5a and 5B should be remaining mass not weight loss.

Answer 8. As per reviewer comment, weight loss  by remaining mass.

9- The caption of Figure 5 must be thermal gravimetric analysis not thermal graphs.

Answer 9. As per reviewer comment, thermal graphs were modified by thermal gravimetric analysis.

10- On page 9, what is the meaning of 2%? is it 2 mg of adsorbent per 100 mg of mixture?

Answer 10. Yes, the meaning of 2% it is 2 mg of adsorbent per 100 mg of mixture.

 11- On Figure ^a, the maximum was achieved at 1% not as mentioned at 2%.

Answer 11. As per reviewer comment, yes this is corrected as: the maximum was achieved at 1%.

12- What is the y-axes of Figure 6a? is it SB content? or mass of adsorbent?

Answer 12. The y-axes of Figure 6a represents the adsorption capacity of methylene blue.

13- Add the experimental contions to the caption of Figure 6.

Answer 13. As per reviewer comment, the experimental conditions are now added to the caption of Figure 6.

14- There is no Figure 6g as mentioned in the caption and in line 289 of page 11.

Answer 14. As per reviewer comment, Figure 6 g is now provided. We are sorry, Figure 6 g was deleted by error upon uploading.

15- on page 16, line 375, it is mentioned that the adsorption was achieved via multilayer, which is usually attributed to physical adsorption not chemical as mentioned in the discussion. to ensure this point is accurate, it is better to calculate the activation energy.

Answer 15. Indeed, the isotherms of Langmuir, Freundlich, and temkim were investigated in this study. The best fitting was selected based on the highest regression coefficient, following the literature.

We Finally thank very much the reviewer for the useful comments and suggestions

Reviewer 2 Report

This manuscript describes the functionalization of pectin with an organometallic compound for designing a sorbent that is applied for methylene blue removal from aqueous solutions. This topic may be of interest for the readers of the journal. The material is widely analyzed using a broad range of analytical procedures.

Editing

(a) Some typing mistakes to be polished: uses and editing of exponents and indices, charcaterization (3.5), etc

(b) Equipment for the characterization of materials is not fully identified (branch, model, location, for example).

(c) Editing of Figure 1: please, correct the editing of [Sb (TPP)Cl2]+ (the “2” index figure has disappeared)

(d) Is it possible revising the sentence at lines 222-223?

(e) Please, clarify the X-axis in Figures 6e and Figure 6f: the Authors report the sorption isotherms (with q plotted vs. Ceq) while in the text they refer to initial dye concentration (Line 274).

(f) Caption of Figure 6 refers to (f-g) panels, while panel g is not reported.

(g) Please, edit homogeneously the references: (a) capital letters in the titles of the papers, (b) abbreviated/non-abbreviated given names of authors, (c) abbreviated/non-abbreviated names of the journals, (d) double numbering of references, etc.

Specific questions

(1) How many replicates were carried out? This should be documented and if the Authors proceeded to duplicate tests, the sorption data should be completed documenting the effective average values with error bars.

(2) Is it possible setting an appropriate reference to support and justify the conclusion on Positive EIS mass spectrum (Figure 1)?
(3) Table 1 provides the XRF analysis of a composite beads; but the Authors did not specify the type of bead (meaning the amount of organometallic compound introduced in the preparation: 0.5%, 1%, or 2%?). This should be clarified. Is there a relationship between the effective amount quantified in the bead and the actual amount introduced in the synthesis procedure?

(4) Determining the pHPZC of the material by titration or through zeta metric measurements would be helpful in the discussion of pH effect.

In addition, the Authors could document the pH variation during the sorption process. In Figure 6b, they do no indicate if this is the initial pH value or the equilibrium value. Actually, this is the equilibrium value that could be preferentially used for evaluating the equilibrium distribution of MB between the two phases.

(5) How were obtained the parameters of modelled curves  (for both kinetics and equilibrium curves): Linear regression (apparently) or non-linear regression analysis? Usually the non-linear regression allows improving the mathematical quality of the fit.

(6) Multi-panel Figures 7 and 8 could probably be re-organized showing only the best model (second order) and forwarding the plots of the other models in Supplementary Information? Same comment for Figure 9 and 10 (with the best model fitting in the core and the other plots in Supplementary Section (especially because Table 3 provides direct information on the quality of fit).

(7) Showing the effective sorption isotherm (q=f(c)) is usually more “instructive” than the plot of fittings. The fact that Freundlich equation best fit experimental data is frequently associated with experimental conditions that do not cover the full range of concentration for reaching the complete saturation of the sorbent. (q plot vs. C would help in showing the saturation – or not). I would suggest the Authors moderating this conclusion concerning this comment (after verification)

(8) The evaluation of the thermodynamic parameters using the method of distribution coefficient (Kd = q/C) presents several disadvantages that were perfectly documented by lima et al. (2019):

E. C. Lima, A. Hosseini-Bandegharaei, J. C. Moreno-Piraján and I. Anastopoulos (2019)

A critical review of the estimation of the thermodynamic parameters on adsorption equilibria. Wrong use of equilibrium constant in the Van't Hoof equation for calculation of thermodynamic parameters of adsorption, Journal of Molecular Liquids, 273, 425-434.

They also suggest using a minimum of four different temperatures for accurate determination of thermodynamic parameters. I would suggest the Authors considering the manuscript for effective calculation of thermodynamic constants:

H. N. Tran, E. C. Lima, R.-S. Juang, J.-C. Bollinger and H.-P. Chao (2021) Thermodynamic parameters of liquid–phase adsorption process calculated from different equilibrium constants related to adsorption isotherms: A comparison study, Journal of Environmental Chemical Engineering, 9(6) 106674.

(9) The evaluation of a new sorbent requires considering the desorption and reuse of the sorbent, especially when the sorbent includes the use of expensive reagents.

(10) The authors do not provide a comparison of the performances of their adsorbent with existing literature. This would help in evaluating the potential of their material. Actually, the sorption performance for MB is relatively weak, and the sorption capacity for pectin is only weakly improved by the functionalization. The comparative table would help in evaluating the competitiveness of this new material.

(11) Is there any risk to release antimony at use and recycling? This may have an environmental impact (antimony, etc.). Could the Authors document this question?

Author Response

Reviewer: 2

This manuscript describes the functionalization of pectin with an organometallic compound for designing a sorbent that is applied for methylene blue removal from aqueous solutions. This topic may be of interest for the readers of the journal. The material is widely analyzed using a broad range of analytical procedures.

Editing

(a) Some typing mistakes to be polished: uses and editing of exponents and indices, charcaterization (3.5), etc

Answer. As per reviewer comment, the manuscript is checked again for such mistake, and every change is done with red color.

(b) Equipment for the characterization of materials is not fully identified (branch, model, location, for example).

Answer. As per reviewer comment, the characterization part is now improved accordingly.

(c) Editing of Figure 1: please, correct the editing of [Sb (TPP)Cl2]+ (the “2” index figure has disappeared)

Answer. As per reviewer comment, the editing of [Sb (TPP)Cl2]+ is now corrected.

(d) Is it possible revising the sentence at lines 222-223?

Answer: As per reviewer comment, the lines 222-223 are now revised as:

The total weight loss, for pectin-[Sb(TPP)Cl2] composite beads, is equal to 71%. This weight loss is subdivided into many stages observed at 8%, 40%, 12%, and 11%.

(e) Please, clarify the X-axis in Figures 6e and Figure 6f: the Authors report the sorption isotherms (with q plotted vs. Ceq) while in the text they refer to initial dye concentration (Line 274).

Answer: Yes the sorption isotherms were plotted with q plotted vs. Ceq. We have deleted the term initial dye concentration in the text.

(f) Caption of Figure 6 refers to (f-g) panels, while panel g is not reported.

Answer. As per reviewer comment, Figure 6 g is now provided. We are sorry, Figure 6 g was deleted by error upon uploading.

(g) Please, edit homogeneously the references: (a) capital letters in the titles of the papers, (b) abbreviated/non-abbreviated given names of authors, (c) abbreviated/non-abbreviated names of the journals, (d) double numbering of references, etc.

Answer. As per reviewer comment, References are checked again and changes are now provided.

Specific questions

(1) How many replicates were carried out? This should be documented and if the Authors proceeded to duplicate tests, the sorption data should be completed documenting the effective average values with error bars.

Answer. As per reviewer comment, error bars are now added to the sorption data.

(2) Is it possible setting an appropriate reference to support and justify the conclusion on Positive EIS mass spectrum (Figure 1)?

Answer. As per reviewer comment, references 20 and 21 are now provided.
(3) Table 1 provides the XRF analysis of a composite beads; but the Authors did not specify the type of bead (meaning the amount of organometallic compound introduced in the preparation: 0.5%, 1%, or 2%?). This should be clarified. Is there a relationship between the effective amount quantified in the bead and the actual amount introduced in the synthesis procedure?

Answer. The missing data is now added for table 1 as: 1%

(4) Determining the pHPZC of the material by titration or through zeta metric measurements would be helpful in the discussion of pH effect.

In addition, the Authors could document the pH variation during the sorption process. In Figure 6b, they do no indicate if this is the initial pH value or the equilibrium value. Actually, this is the equilibrium value that could be preferentially used for evaluating the equilibrium distribution of MB between the two phases.

Answer. In the current investigation, we have studied the change of the initial pH on the adsorption capacity in the presence of pectin and composite beads.

(5) How were obtained the parameters of modelled curves  (for both kinetics and equilibrium curves): Linear regression (apparently) or non-linear regression analysis? Usually the non-linear regression allows improving the mathematical quality of the fit.

Answer. Yes, In the current investigation the parameters of modelled curves for both kinetics and equilibrium curves were obtained using Linear regression analysis.

(6) Multi-panel Figures 7 and 8 could probably be re-organized showing only the best model (second order) and forwarding the plots of the other models in Supplementary Information? Same comment for Figure 9 and 10 (with the best model fitting in the core and the other plots in Supplementary Section (especially because Table 3 provides direct information on the quality of fit).

Answer.  As per reviewer comment, Figures 7 and 8 are now re-organized showing only the best model (second order) and we have forwarded the plots of the other models in Supplementary Information. The same comment for Figure 9 and 10 (with the best model fitting in the core and the other plots in Supplementary Section

Figure 7.  Plots of second order kinetic model for: (a) pectin beads, and (b) Pectin-[Sb(TPP)Cl2] beads (1%)

Figure 8.  Plots of Freundlich isotherm for: (a) pectin beads, and (b) pectin-[Sb(TPP)Cl2] beads (1%)

Figure S1. Kinetic plots for pectin beads: (a) First order, (b) Elovich, and (c) Intra-particular diffusion.

Figure S2. Kinetic plots for pectin-[Sb(TPP)Cl2] beads (1%): (a) First order, (b) Elovich, and (c) Intra-particular diffusion.

Figure S3. Isotherm data modeling for pectin beads: (a) Langmuir, and (b) Temkin

Figure S4. Isotherm data modeling for pectin-[Sb(TPP)Cl2] beads (1%): (a) Langmuir, and (b) Temkin                    

(7) Showing the effective sorption isotherm (q=f(c)) is usually more “instructive” than the plot of fittings. The fact that Freundlich equation best fit experimental data is frequently associated with experimental conditions that do not cover the full range of concentration for reaching the complete saturation of the sorbent. (q plot vs. C would help in showing the saturation – or not). I would suggest the Authors moderating this conclusion concerning this comment (after verification)

Answer.  Yes we have studied the sorption isotherm (q as a function of C). We noted that our results indicated that the lowest values of the regression coefficients are detected for the isotherms of Temkin and Langmuir (Figures S2, S3) as compared to those of Freundlich (R2 ≥ 0.98) (Figure 8). This suggests that multilayer adsorption may occur at heterogeneous interface [38].

(8) The evaluation of the thermodynamic parameters using the method of distribution coefficient (Kd = q/C) presents several disadvantages that were perfectly documented by lima et al. (2019):

  1. C. Lima, A. Hosseini-Bandegharaei, J. C. Moreno-Piraján and I. Anastopoulos (2019)

A critical review of the estimation of the thermodynamic parameters on adsorption equilibria. Wrong use of equilibrium constant in the Van't Hoof equation for calculation of thermodynamic parameters of adsorption, Journal of Molecular Liquids, 273, 425-434.

They also suggest using a minimum of four different temperatures for accurate determination of thermodynamic parameters. I would suggest the Authors considering the manuscript for effective calculation of thermodynamic constants:

  1. N. Tran, E. C. Lima, R.-S. Juang, J.-C. Bollinger and H.-P. Chao (2021) Thermodynamic parameters of liquid–phase adsorption process calculated from different equilibrium constants related to adsorption isotherms: A comparison study, Journal of Environmental Chemical Engineering, 9(6) 106674.

Answer. In our study we have used the following formula: Kd = qe/Ce to calculate the thermodynamic constants.

(9) The evaluation of a new sorbent requires considering the desorption and reuse of the sorbent, especially when the sorbent includes the use of expensive reagents.

Answer.  Yes, the desorption experiments could be studied. This certainly needs many experiments to search for the best agent useful to desorb methylene blue from the prepared beads.

(10) The authors do not provide a comparison of the performances of their adsorbent with existing literature. This would help in evaluating the potential of their material. Actually, the sorption performance for MB is relatively weak, and the sorption capacity for pectin is only weakly improved by the functionalization. The comparative table would help in evaluating the competitiveness of this new material.

Answer. As per reviewer comment, a table is now added (Table 2) to compare the performances of the prepared actual adsorbents with existing literature.

Table 2. Comparison of the maximum adsorption capacities of methylene blue with some examples of adsorbents reported in the literature

Adsorbent

Maximum adsorption capacity (mg/g)

Reference

Pectin-[Sb(TPP)Cl2] composite beads (1%)

68

Current work

Graphene oxide@chitosan composite beads

23.2

[36]

Cellulose–chitosan composite beads 

55

[37]

ZnO-chitosan nanocomposites

98

[38]

Chitosan–montmorillonite/polyaniline nanocomposite

111

[39]

Sodium alginate-hydroxyapatite composite

142

[40]

(11) Is there any risk to release antimony at use and recycling? This may have an environmental impact (antimony, etc.). Could the Authors document this question?

Answer. Based on our previous investigations, we did not believe the release of the metal from the beads.

We Finally thank very much the reviewer for the useful comments and suggestions

Round 2

Reviewer 2 Report

I thank the Reviewers for having at least partly answered to my questions and made some appropriate modifications. In some (few) items I do not share the opinion of the Authors who only comment the technique or methodology they used without changing the content. I respect the "diversity" in opinion and do not consider that the paper requires fundamental additional changes. However, i would like to suggest the Authors to consider the suggestions for future work.

Please note that there is probably a confusion in the editing of the authors in ref. 25.

Author Response

Reviewer: 2

I thank the Reviewers for having at least partly answered to my questions and made some appropriate modifications. In some (few) items I do not share the opinion of the Authors who only comment the technique or methodology they used without changing the content. I respect the "diversity" in opinion and do not consider that the paper requires fundamental additional changes. However, i would like to suggest the Authors to consider the suggestions for future work.

Please note that there is probably a confusion in the editing of the authors in ref. 25.

Answer. As per reviewer comment, the reference 25 is now corrected as:

  1. Raoudha, S.; Marwa, C.; Mahjoub, J.; Tawfik, A. S.; Rafik, B.C.; Eric, S-A.; Frederique, L.; Christain, P.; Abdul-Rahman, A.; Habib, N. Meso-tetrakis (3, 4, 5-trimethoxyphenyl) porphyrin derivatives: synthesis, spectroscopic characterizations and adsorption of NO2, Chem. Eng. J. 2019, 375, 122005.